# A Multilinear Least-Squares Formulation for Sparse Tensor Canonical Correlation Analysis

**Jun Yu** *juy220@lehigh.edu*
*Department of Computer Science and Engineering*
*Lehigh University*

**Zhaoming Kong** *kong.zm@mail.scut.edu.cn*
*School of Software Engineering*
*South China University of Technology*

**Kun Chen** *kun.chen@uconn.edu*
*Department of Statistics*
*University of Connecticut*

**Xin Zhang** *henry@stat.fsu.edu*
*Department of Statistics*
*Florida State University*

**Yong Chen** *ychen123@pennmedicine.upenn.edu*
*Department of Biostatistics, Epidemiology and Informatics*
*University of Pennsylvania*

**Lifang He** *lih319@lehigh.edu*
*Department of Computer Science and Engineering*
*Lehigh University*

**Reviewed on OpenReview:** *https://openreview.net/forum?id=zcOYOcAuTV*

## Abstract

Tensor data are becoming important recently in various applications, e.g., image and video recognition, which pose new challenges for data modeling and analysis approaches, such as high-order relations of large complexity, varying data scale and gross noise. In this paper, we consider the problem of sparse canonical correlation analysis for arbitrary tensor data. Although several methods have been proposed for this task, there are still limitations hindering its practical applications. To this end, we present a **g**eneral **S**parse **T**ensor **C**anonical **C**orrelation **A**nalysis (gSTCCA) method from a multilinear least-squares perspective. Specifically, we formulate the problem as a constrained multilinear least-squares problem with tensor-structured sparsity regularization based on CANDECOMP/PARAFAC (CP) decomposition. Then we present a divide-and-conquer deflation approach to tackle the problem by successive rank-one tensor estimation of the residual tensors, where the overall model is broken up into a set of unconstrained linear least-squares problems that can be efficiently solved. Through extensive experiments conducted on five different datasets for recognition tasks, we demonstrate that the proposed method achieves promising performance compared to the SOTA vector- and tensor-based canonical correlation analysis methods in terms of classification accuracy, model sparsity, and robustness to missing and noisy data. The code is publicly available at https://github.com/junfish/gSTCCA.

## 1 Introduction

Canonical Correlation Analysis (CCA) (Hotelling, 1936) is a powerful tool for correlation analysis and dimensionality reduction that has been successfully applied in a wide spectrum of fields, such as industrial process (Zhu et al., 2023), neuroscience (Mihalik et al., 2022), speech processing (Choukri & Chollet, 1986), natural language processing (Dhillon et al., 2011), computer vision (Lu et al., 2022), and graph learning (Zhang et al., 2021). Suppose that two sets of measurements are available on the same set of observations, the classical

CCA seeks linear combinations or loadings of all variables in each set that are maximally correlated with each other. Over the years, CCA has been extensively studied from multiple perspectives (Hardoon et al., 2004; Yang et al., 2019; Lindenbaum et al., 2021; Sanghavi & Verma, 2022; Friedlander & Wolf, 2023). In particular, CCA provides fully dense loadings, it can often lead to difficulty in interpretation of results and disturbance from noisy features (Park & Konishi, 2020). When the number of variables far exceeds the number of subjects, classical CCA methods are no longer appropriate (Chalise & Fridley, 2012). Therefore, a variety of models have been proposed for sparse CCA (Chu et al., 2013; Wilms & Croux, 2015; Gao et al., 2017). Furthermore, the sparsity of model weights provides essential interpretability in numerous applications (Witten et al., 2009; Qiu et al., 2022). Nevertheless, these approaches have been limited to vector-type data and cannot effectively deal with high order tensor data.

In many real-world applications, data are frequently organized in the form of tensors. Specifically, a grayscale face image is transformed as a second-order tensor of *height* × *width* and a gait sequence is arranged as a third-order tensor of *body height* × *body width* × *gait length*. A common approach for applying CCA methods to tensor data is to convert it into vectors by flattening the tensor element by element. However, such an approach ignores the latent higher-order structure of pixel arrangement, such as spatial coherence. Additionally, the high dimensionality of the resulting vectors can lead to computational challenges and inefficiency (Zhou et al., 2013).

In recent years, there has been increasing interests in generalizing CCA methods to tensors. Several tensor-based methods have been proposed to overcome the above limitations (Chen et al., 2021; Gang et al., 2011; Kim et al., 2007b; Lee & Choi, 2007; Min et al., 2019; Wang et al., 2016). The objective is to discover relationships between two tensors while simultaneously preserving multidimensional structure of the tensors and utilizing substantially fewer parameters. While these methods have shown promising results in several applications, most of them directly maximize the correlation between two tensor spaces and require input tensors having the same order or even the same size, which are not practical for general use. Furthermore, existing sparse tensor approaches (Min et al., 2019; Wang et al., 2016) focus on quantifying the sparsity of factor vectors, where the high-order correlations between different modes of tensors are ignored. The use of structured sparsity in tensors is still an issue under the constraint of unit variance in CCA criteria.

In this paper, we propose a general sparse tensor CCA (gSTCCA) method for analyzing paired tensor data with arbitrary order, which exploits the tensor structure to encourage tensor-level sparsity. The main contributions of this paper are threefold:

- We formulate the gSTCCA model as a constrained multilinear least-squares regression problem with tensor-structured sparsity regularization based on the CANDECOMP/PARAFAC (CP) decomposition, which is both parsimonious and highly interpretable.

- We present an efficient divide-and-conquer approach to solve gSTCCA based on successive rank-one tensor estimation of the residual tensors, and theoretically show that the whole problem boils down to a set of unconstrained alternating least-squares problems.

- We conduct extensive experiments on five real-world classification datasets with varying tensor sizes and orders. The results reveal that gSTCCA outperforms SOTA vector- and tensor-based CCA methods in terms of accuracy, model sparsity, and robustness to missing and noisy data. Moreover, gSTCCA demonstrates noteworthy self-supervised capabilities within the framework of CCA, adapting effectively to downstream classification tasks.

## 2 Related Works

In this section, we briefly review the existing works that are relevant to our approach.

**Vector-based CCA Approaches.** CCA has been extensively studied in vector space from many aspects. For example, a number of sparse CCA variants (Chu et al., 2013; Gao et al., 2017; Wilms & Croux, 2015; Witten & Tibshirani, 2009) have been proposed to impose sparsity constraints on the canonical loading vectors. In particular, the $\ell_1$-norm regularization is commonly used for its convexity and ease of optimization.

Various nonlinear methods have been proposed to capture nonlinear dependency in the data based on kernel methods (Hardoon et al., 2004; Zheng et al., 2006) and deep learning methods (Andrew et al., 2013; Yang et al., 2017). Some supervised approaches (Arandjelović, 2014; Kim et al., 2007a) have been proposed to exploit label information by adding discriminative term to the classical CCA. Several multi-set CCA methods (Correa et al., 2009; 2010; Parra, 2018) have been proposed to integrate multiple datasets. In addition, it has been shown that under a mild condition, CCA can be formulated as a least-squares problem (Sun et al., 2008).

**Tensor-based CCA Approaches.** Several tensor-based CCA methods have recently been proposed for tensor data. For example, Lee & Choi (2007) proposed two-dimensional CCA (2DCCA) via low-rank matrix factorization. Gang et al. (2011) extended the 2DCCA to 3DCCA. Kim et al. (2007b) designed a shared-mode tensor CCA (TCCA) model to capture the relations among tensor modes. Chen et al. (2021) applied CP decomposition and devised a deflation-based model (dTCCA) using the higher-order power method. Yan et al. (2011) proposed sparse 2DCCA (S2DCCA) with $\ell_1$-norm regularization. Wang et al. (2016) generalized S2DCCA to higher-order tensors and proposed sparse tensor CCA (STCCA) by iteratively seeking CCA components for each tensor mode. Min et al. (2019) proposed a population model via CP decomposition and adapted it to the sparse case with the hard-thresholding technique called the spTCCA.

Generally, most of the above tensor methods focus on extending the classical CCA to maximize the correlation coefficient between tensors, and require input tensors having the same order or size. Chen et al. (2021) solved the problem from multilinear least-squares perspective, which is the most closely related work to us, however, it is confined to tensor data of the same order and is not a sparse method. Furthermore, existing sparse tensor models (Min et al., 2019; Wang et al., 2016) focus on inducing the sparsity of the factor vectors, there still lacks a general and effective model that can directly learn tensor-level sparsity.

## 3 Preliminaries

**Notation.** Tensors are denoted by bold calligraphic letters, e.g., $\mathcal{A} \in \mathbb{R}^{I_1 \times \cdots \times I_N}$. Matrices are denoted by bold uppercase letters, e.g., $\mathbf{A} \in \mathbb{R}^{I_1 \times I_2}$. Vectors are denoted by bold lowercase letters, e.g., $\mathbf{a} \in \mathbb{R}^I$. Indices are denoted by lowercase letters spanning the range from 1 to the uppercase letter of the index, e.g., $n = 1, \cdots, N$. We denote the entries by $a_i$, $a_{i,j}$, $a_{i,j,k}$, etc., depending on the number of dimensions. Thus, each entry of an $N$th-order tensor $\mathcal{A} \in \mathbb{R}^{I_1 \times \cdots \times I_N}$ is indexed by $N$ indices $\{i_n\}_{n=1}^N$, and each $i_n$ indexes the $n$-mode of $\mathcal{A}$. Specifically, $-n$ denotes every mode except $n$, $Vec(\cdot)$ denotes the vectorization operator, and $Tr(\cdot)$ denotes the trace of a matrix.

### 3.1 Tensor Algebra

We refer readers to Kolda & Bader (2009) for more details of tensor algebra.

**Definition 3.1** (Inner product). The inner product of two tensors $\mathcal{A}, \mathcal{B} \in \mathbb{R}^{I_1 \times \cdots \times I_N}$ is the sum of the products of their entries, defined as $\langle \mathcal{A}, \mathcal{B} \rangle = \sum_{i_1=1}^{I_1} \cdots \sum_{i_1=1}^{I_N} a_{i_1, \cdots, i_N} b_{i_1, \cdots, i_N}$.

It follows immediately that the $\ell_2$ norm of $\mathcal{A}$ is defined as $\|\mathcal{A}\|_2 = \sqrt{\langle \mathcal{A}, \mathcal{A} \rangle}$. The $\ell_1$ norm of a tensor is defined as $\|\mathcal{A}\|_1 = \sum_{i_1=1}^{I_1} \cdots \sum_{i_N=1}^{I_N} |a_{i_1, \cdots, i_N}|$.

**Definition 3.2** (Tensor product). Let $\mathbf{a}^{(n)} \in \mathbb{R}^{I_n}$ be a vector with dimension $I_n$. The tensor product of $N$ vectors, denoted by $\mathcal{A} = \mathbf{a}^{(1)} \circ \cdots \circ \mathbf{a}^{(N)}$, is an $(I_1 \times \cdots \times I_N)$-tensor of which the entries are given by $\mathcal{A}_{i_1, \cdots, i_N} = a_{i_1}^{(1)} \cdots a_{i_N}^{(N)}$. We call $\mathcal{A}$ a rank-one tensor or a unit-rank tensor.

**Definition 3.3** ($n$-mode product). The $n$-mode product of a tensor $\mathcal{A} \in \mathbb{R}^{I_1 \times \cdots \times I_N}$ by a vector $\mathbf{u} \in \mathbb{R}^{I_n}$, denoted by $\mathcal{A} \times_n \mathbf{u}$, is an $(I_1 \times \cdots \times I_{n-1} \times I_{n+1} \cdots \times I_N)$-tensor of which the entries are given by $(\mathcal{A} \times_n \mathbf{u})_{i_1, \ldots, i_{n-1} i_{n+1}, \ldots, i_N} = \sum_{i_n=1}^{I_n} a_{i_1, \ldots, i_N} u_{i_n}$.

**Definition 3.4** (CP decomposition). For any tensor $\mathcal{A} \in \mathbb{R}^{I_1 \times \cdots \times I_N}$, the CP decomposition is defined as $\mathcal{A} = \sum_{r=1}^R \sigma_r \mathbf{a}_r^{(1)} \circ \cdots \circ \mathbf{a}_r^{(N)}$, where $\sigma_r \in \mathbb{R}$, $\mathbf{a}_r^{(n)} \in \mathbb{R}^{I_n}$ and $\|\mathbf{a}_r^{(n)}\|_2 = 1$, and $\mathbf{A}^{(n)} = [\mathbf{a}_1^{(n)}, \cdots, \mathbf{a}_R^{(n)}]$ for $n = 1, \cdots, N$ are called factor matrices.

**Definition 3.5** (Tensor rank[1]). The tensor rank of $\mathcal{A}$, denoted by $Rank(\mathcal{A})$, is the smallest number $R$ such that the CP decomposition is exact.

**Definition 3.6** (Orthogonality (Kolda, 2001)). Two rank-one tensors $\mathcal{A}_i$ and $\mathcal{A}_j$ are orthogonal ($\mathcal{A}_i \perp \mathcal{A}_j$), iff

$$\langle \mathcal{A}_i, \mathcal{A}_j \rangle = \prod_{n=1}^{N} \langle \mathbf{a}_i^{(n)}, \mathbf{a}_j^{(n)} \rangle = 0, \tag{1}$$

where $\mathbf{a}_i^{(n)}$ and $\mathbf{a}_j^{(n)}$ are unit vectors based on CP.

### 3.2 Canonical Correlation Analysis

Let $\mathbf{X} \in \mathbb{R}^{P \times N}$ and $\mathbf{Y} \in \mathbb{R}^{Q \times N}$ be two sets of $N$ paired observations. Without loss of generality, assume both $\{\mathbf{x}_n\}_{n=1}^{N}$ and $\{\mathbf{y}_n\}_{n=1}^{N}$ have zero mean, i.e., $\sum_{n=1}^{N} \mathbf{x}_n = \mathbf{0}$ and $\sum_{n=1}^{N} \mathbf{y}_n = \mathbf{0}$. CCA aims to maximize the correlation between the projections of two sets by

$$\max_{\mathbf{u},\mathbf{v}} \mathbf{u}^{\mathrm{T}} \mathbf{X} \mathbf{Y}^{\mathrm{T}} \mathbf{v}, \text{ s.t. } \mathbf{u}^{\mathrm{T}} \mathbf{X} \mathbf{X}^{\mathrm{T}} \mathbf{u} = \mathbf{v}^{\mathrm{T}} \mathbf{Y} \mathbf{Y}^{\mathrm{T}} \mathbf{v} = N, \tag{2}$$

to get the first pair of *canonical vectors* $\mathbf{u} \in \mathbb{R}^{P}$ and $\mathbf{v} \in \mathbb{R}^{Q}$, which are further used to obtain the first pair of *canonical variates* $\mathbf{u}^{\mathrm{T}}\mathbf{X}$ and $\mathbf{v}^{\mathrm{T}}\mathbf{Y}$. To obtain multiple projections of CCA, we can recursively solve the above optimization problem with additional constraint that the current canonical variates must be orthogonal to all previous ones (Chu et al., 2013). Specifically, denoting $\mathbf{U} = [\mathbf{u}_1, \cdots, \mathbf{u}_S] \in \mathbb{R}^{P \times S}$ and $\mathbf{V} = [\mathbf{v}_1, \cdots, \mathbf{v}_S] \in \mathbb{R}^{Q \times S}$, Eq. (2) can be reformulated to get multiple projections as

$$\max_{\mathbf{U},\mathbf{V}} Tr(\mathbf{U}^{\mathrm{T}} \mathbf{X} \mathbf{Y}^{\mathrm{T}} \mathbf{V}), \quad \text{s.t. } \mathbf{U}^{\mathrm{T}} \mathbf{X} \mathbf{X}^{\mathrm{T}} \mathbf{U} = \mathbf{V}^{\mathrm{T}} \mathbf{Y} \mathbf{Y}^{\mathrm{T}} \mathbf{V} = \mathbf{I}_S. \tag{3}$$

One appealing property of CCA is that it can be formulated from the view of least-squares problems as follows (Brillinger, 1975; Chu et al., 2013):

$$\min_{\mathbf{U},\mathbf{V}} \|\mathbf{U}^{\mathrm{T}} \mathbf{X} - \mathbf{V}^{\mathrm{T}} \mathbf{Y}\|_2^2, \quad \text{s.t. } \mathbf{U}^{\mathrm{T}} \mathbf{X} \mathbf{X}^{\mathrm{T}} \mathbf{U} = \mathbf{V}^{\mathrm{T}} \mathbf{Y} \mathbf{Y}^{\mathrm{T}} \mathbf{V} = \mathbf{I}_S. \tag{4}$$

Once we obtain canonical matrices $\mathbf{U}$ and $\mathbf{V}$, the projection of $\mathbf{X}$ and $\mathbf{Y}$ into the new space can be obtained by $(\mathbf{U}^{\mathrm{T}}\mathbf{X}, \mathbf{V}^{\mathrm{T}}\mathbf{Y})$.

## 4 Sparse Tensor Canonical Correlation Analysis

To illustrate our design, we start with the first pair of canonical rank-one tensors, and then present a deflation method to find more than one canonical rank-one component.

### 4.1 Model Formulation

Given $N$ paired samples of tensors $\{\mathcal{X}_n, \mathcal{Y}_n\}_{n=1}^{N}$, where $\mathcal{X}_n \in \mathbb{R}^{P_1 \times \cdots \times P_S}$ and $\mathcal{Y}_n \in \mathbb{R}^{Q_1 \times \cdots \times Q_T}$. Without loss of generality, we assume that $\{\mathcal{X}_n\}_{n=1}^{N}$ and $\{\mathcal{Y}_n\}_{n=1}^{N}$ are normalized to have mean zero and standard deviation one on each feature. We consider the STCCA model of the form

$$\min_{\mathcal{U},\mathcal{V}} \frac{1}{2N} \sum_{n=1}^{N} (\langle \mathcal{U}, \mathcal{X}_n \rangle - \langle \mathcal{V}, \mathcal{Y}_n \rangle)^2 + \lambda_u \|\mathcal{U}\|_1 + \lambda_v \|\mathcal{V}\|_1,$$

$$\text{s.t. } \frac{1}{N} \sum_{n=1}^{N} \langle \mathcal{U}, \mathcal{X}_n \rangle^2 = 1, \frac{1}{N} \sum_{n=1}^{N} \langle \mathcal{V}, \mathcal{Y}_n \rangle^2 = 1, Rank(\mathcal{U}) = Rank(\mathcal{V}) = 1, \tag{5}$$

where the low-rank constraints are enforced through the canonical tensors $\mathcal{U}$ and $\mathcal{V}$ in order to reduce the complexity of the model and leverage the structural information, and the tensor-structured $\ell_1$ norm is employed to encourage sparsity in the multilinear combination of decision variables.

---

[1]There are multiple ways to define the tensor rank. In this paper, we define it based on the CP decomposition and call it CP-rank.

Based on the definition of tensor rank, we assume that $\mathcal{U} = \sigma_u \mathbf{u}^{(1)} \circ \cdots \circ \mathbf{u}^{(S)}, \mathcal{V} = \sigma_v \mathbf{v}^{(1)} \circ \cdots \circ \mathbf{v}^{(T)}$, where $\sigma_u, \sigma_v > 0$, $\|\mathbf{u}^{(s)}\|_1 = 1$ and $\|\mathbf{v}^{(t)}\|_1 = 1$. Note that this amounts to using an $\ell_1$ normalization, instead of the $\ell_2$ norm preferred by mathematicians, and the trade-off is in the simplicity of formulae. $\sigma_u \geq 0$ and $\sigma_v > 0$ can always hold by flipping the signs of the factors.

It becomes evident that $\|\mathcal{U}\|_1 = \sigma_u \prod_{s=1}^{S} \|\mathbf{u}^{(s)}\|_1$ and $\|\mathcal{V}\|_1 = \sigma_v \prod_{t=1}^{T} \|\mathbf{v}^{(t)}\|_1$. In other words, the sparsity of a rank-one tensor directly leads to the sparsity of its components. This allows us to kill multiple birds with one stone: by simply pursuing the element-wise sparsity of the canonical rank-one tensor with only one tunable parameter $\lambda$ in each view, solving (5) can produce a set of joint sparse factors $\mathbf{u}^{(s)}$ for $s = 1, \cdots, S$ simultaneously (same as $\mathbf{v}^{(t)}$, $t = 1, \cdots, T$).

Continuing with the unified representation, let $\mathcal{X} = [\mathcal{X}_1, \cdots, \mathcal{X}_N] \in \mathbb{R}^{P_1 \times \cdots \times P_S \times N}$ and $\mathcal{Y} = [\mathcal{Y}_1, \cdots, \mathcal{Y}_N] \in \mathbb{R}^{Q_1 \times \cdots \times Q_T \times N}$. We denote by $\mathcal{X} \times_1^S \mathcal{U} = \mathcal{X} \times_1 \sigma_u \mathbf{u}^{(1)} \cdots \times_S \mathbf{u}^{(S)}$ a $N$-dimension vector with $n$-th element $\langle \mathcal{U}, \mathcal{X}_n \rangle$, and $\mathcal{Y} \times_1^T \mathcal{V} = \mathcal{Y} \times_1 \sigma_v \mathbf{v}^{(1)} \cdots \times_T \mathbf{v}^{(T)}$ a $N$-dimension vector with $n$-th element $\langle \mathcal{V}, \mathcal{Y}_n \rangle$. Eq. (5) can be written in a unified compact form as

$$\min_{\mathcal{U}, \mathcal{V}} \frac{1}{2N} \|\mathcal{X} \times_1^S \mathcal{U} - \mathcal{Y} \times_1^T \mathcal{V}\|_2^2 + \lambda_u \|\mathcal{U}\|_1 + \lambda_v \|\mathcal{V}\|_1,$$
$$\text{s.t. } Var(\mathcal{X} \times_1^S \mathcal{U}) = Var(\mathcal{Y} \times_1^T \mathcal{V}) = 1, Rank(\mathcal{U}) = Rank(\mathcal{V}) = 1. \tag{6}$$

### 4.2 Rank-One Estimation

The objective function in Eq. (6) is a non-convex multi-objective optimization problem, which makes it hard to solve. Inspired by He et al. (2018), we show that this problem can be broken up into a set of unconstrained alternating least-squares problems which can be efficiently solved.

Suppose we have an initial value $\mathcal{V}^*$ for the canonical tensor with respect to $\mathcal{V}$. By fixing it, the optimization problem in (6) reduces to

$$\widehat{\mathcal{U}}|\mathcal{V}^* = \arg\min_{\mathcal{U}} \frac{1}{2N} \|\mathcal{X} \times_1^S \mathcal{U} - \mathcal{Y} \times_1^T \mathcal{V}^*\|_2^2 + \lambda_u \|\mathcal{U}\|_1,$$
$$\text{s.t. } Var(\mathcal{X} \times_1^S \mathcal{U}) = 1, \ Rank(\mathcal{U}) = 1. \tag{7}$$

Similarly, for a fixed value $\mathcal{U}^*$, the optimal value for $\mathcal{V}$ can also be reduced to

$$\widehat{\mathcal{V}}|\mathcal{U}^* = \arg\min_{\mathcal{V}} \frac{1}{2N} \|\mathcal{X} \times_1^S \mathcal{U}^* - \mathcal{Y} \times_1^T \mathcal{V}\|_2^2 + \lambda_v \|\mathcal{V}\|_1,$$
$$\text{s.t. } Var(\mathcal{Y} \times_1^T \mathcal{V}) = 1, Rank(\mathcal{V}) = 1. \tag{8}$$

Let us now for a moment assume that we will solve Eqs. (7) and (8) by the alternating least squares (ALS) algorithm. We alternately optimize with respect to variables $(\sigma_u, \mathbf{u}^{(s)})$ or $(\sigma_v, \mathbf{v}^{(t)})$ with others fixed. Let $\widehat{\mathbf{u}}^{(s)} = \sigma_u \mathbf{u}^{(s)}$, $\mathbf{X}^{(-s)} = \mathcal{X} \times_1 \mathbf{u}^{(1)} \times_2 \cdots \times_{s-1} \mathbf{u}^{(s-1)} \times_{s+1} \cdots \times_S \mathbf{u}^{(S)}$, and let $\widehat{\mathbf{v}}^{(t)} = \sigma_v \mathbf{v}^{(t)}$, $\mathbf{Y}^{(-t)} = \mathcal{Y} \times_1 \mathbf{v}^{(1)} \times_2 \cdots \times_{t-1} \mathbf{v}^{(t-1)} \times_{t+1} \cdots \times_T \mathbf{v}^{(T)}$. The subproblems in (7) with respect to $(\sigma_u, \mathbf{u}^{(s)})$ and $(\sigma_v, \mathbf{v}^{(t)})$ boil down to

$$\min_{\widehat{\mathbf{u}}^{(s)}} \frac{1}{2N} \|\widehat{\mathbf{u}}^{(s)\text{T}} \mathbf{X}^{(-s)} - \mathbf{v}^{(t)*\text{T}} \mathbf{Y}^{(-t)}\|_2^2 + \lambda_u \|\widehat{\mathbf{u}}^{(s)}\|_1, \quad \text{s.t. } Var(\widehat{\mathbf{u}}^{(s)\text{T}} \mathbf{X}^{(-s)}) = 1, \tag{9}$$

$$\min_{\widehat{\mathbf{v}}^{(t)}} \frac{1}{2N} \|\mathbf{u}^{(s)*\text{T}} \mathbf{X}^{(-s)} - \widehat{\mathbf{v}}^{(t)\text{T}} \mathbf{Y}^{(-t)}\|_2^2 + \lambda_v \|\widehat{\mathbf{v}}^{(t)}\|_1, \quad \text{s.t. } Var(\widehat{\mathbf{v}}^{(t)\text{T}} \mathbf{Y}^{(-t)}) = 1. \tag{10}$$

Once we obtain the solution $\widehat{\mathbf{u}}^{(s)}$, we can set $\sigma_u = \|\widehat{\mathbf{u}}^{(s)}\|_1$ and $\mathbf{u}^{(s)} = \widehat{\mathbf{u}}^{(s)}/\sigma_u$ to satisfy the constraints whenever $\widehat{\mathbf{u}}^{(s)} \neq \mathbf{0}$. If $\widehat{\mathbf{u}}^{(s)} = \mathbf{0}$, $\mathbf{u}^{(s)}$ is no longer identifiable, we then set $\sigma_u = 0$ and terminate the algorithm. Analogously, $\sigma_v$ and $\mathbf{v}^{(t)}$ can be computed.

Inspired by the Lemma 3 in Mai & Zhang (2019), we present the following Lemma 4.1. We accordingly get rid of the unit variance constraints in Eqs. (9) and (10).

**Lemma 4.1.** *Consider the following minimization problem*

$$\min_{\widehat{\mathbf{u}}} \frac{1}{2N}\|\widehat{\mathbf{u}}^{\mathrm{T}}\mathbf{X} - \mathbf{y}\|_2^2 + \lambda_u\|\widehat{\mathbf{u}}\|_1, \quad s.t.\ Var(\widehat{\mathbf{u}}^{\mathrm{T}}\mathbf{X}) = \frac{1}{N}\widehat{\mathbf{u}}^{\mathrm{T}}\mathbf{X}\mathbf{X}^{\mathrm{T}}\widehat{\mathbf{u}} = 1, \tag{11}$$

*where $\mathbf{X}$ and $\mathbf{y}$ are fixed, and $\lambda_u$ is a hyperparameter that controls sparsity. The solution to Eq. (11) is $\widehat{\mathbf{u}}^* = (Var(\widetilde{\mathbf{u}}^{*\mathrm{T}}\mathbf{X}))^{-1/2}\widetilde{\mathbf{u}}^*$, where $\widetilde{\mathbf{u}}^*$ is obtained via*

$$\widetilde{\mathbf{u}}^* = \arg\min_{\widetilde{\mathbf{u}}} \frac{1}{2N}\|\widetilde{\mathbf{u}}^{\mathrm{T}}\mathbf{X} - \mathbf{y}\|_2^2 + \lambda_u\|\widetilde{\mathbf{u}}\|_1 \tag{12}$$

The proof of **Lemma** 4.1 above is shown in Mai & Zhang (2019). Specifically, the solution to Eq. (9) is $\widehat{\mathbf{u}}^{(s)} = (Var(\widetilde{\mathbf{u}}^{(s)\mathrm{T}}\mathbf{X}^{(-s)}))^{-1/2}\widetilde{\mathbf{u}}^{(s)\mathrm{T}}$ with a fixed $\mathbf{v}^{(t)*}$, where $\widetilde{\mathbf{u}}^{(s)}$ is obtained by

$$\min_{\widetilde{\mathbf{u}}^{(s)}} \frac{1}{2N}\|\widetilde{\mathbf{u}}^{(s)\mathrm{T}}\mathbf{X}^{(-s)} - \mathbf{v}^{(t)*\mathrm{T}}\mathbf{Y}^{(-t)}\|_2^2 + \lambda_u\|\widetilde{\mathbf{u}}^{(s)}\|_1. \tag{13}$$

Similarly, the solution to Eq. (10) is $\widehat{\mathbf{v}}^{(t)} = (Var(\widetilde{\mathbf{v}}^{(t)\mathrm{T}}\mathbf{Y}^{(-t)}))^{-1/2}\widetilde{\mathbf{v}}^{(t)\mathrm{T}}$ with a fixed $\mathbf{u}^{(s)*}$, where $\widetilde{\mathbf{v}}^{(t)}$ is obtained by

$$\min_{\widetilde{\mathbf{v}}^{(t)}} \frac{1}{2N}\|\mathbf{u}^{(s)*\mathrm{T}}\mathbf{X}^{(-s)} - \widetilde{\mathbf{v}}^{(t)\mathrm{T}}\mathbf{Y}^{(-t)}\|_2^2 + \lambda_v\|\widetilde{\mathbf{v}}^{(t)}\|_1. \tag{14}$$

Notice that the problem boils down to alternatively solving Eqs. (13) and (14), which are the standard least-squares lasso problems. Here we employ the fast SURF algorithm proposed in He et al. (2018) to solve it, which can trace the solution paths for $\lambda_u$ and $\lambda_v$, in the manner similar to that of stagewise solution for sparse CCA (Sun et al., 2008). Algorithm 1 summarizes the procedure.

---

**Algorithm 1** Rank-One Estimation

**Input:** Paired tensor dataset $\{\mathcal{X}_n, \mathcal{Y}_n\}_{n=1}^N$, step size $\epsilon$.
**Output:** Canonical rank-one tensor pair $\widehat{\mathcal{U}}$ and $\widehat{\mathcal{V}}$.
1: Initialize $\mathcal{U}^*, \mathcal{V}^*$.
2: Set $k = 0$.
3: **while** not converged **do**
4:    Run **SURF**$(\epsilon)$ algorithm to get $\widetilde{\mathcal{U}}_k$ and $\lambda_{u,k}$ by using Eq. (13).
5:    Run **SURF**$(\epsilon)$ algorithm to get $\widetilde{\mathcal{V}}_k$ and $\lambda_{v,k}$ by using Eq. (14).
6:    $k \leftarrow k + 1$.
7: **end while**
8: Compute $\widehat{\mathbf{u}}^{(s)} = (Var(\widetilde{\mathbf{u}}_k^{(s)\mathrm{T}}\mathbf{X}^{(-s)}))^{-\frac{1}{2}}\widetilde{\mathbf{u}}_k^{(s)\mathrm{T}}$, $s = 1, \cdots, S$,
9: Compute $\widehat{\mathbf{v}}^{(t)} = (Var(\widetilde{\mathbf{v}}_k^{(t)\mathrm{T}}\mathbf{Y}^{(-t)}))^{-\frac{1}{2}}\widetilde{\mathbf{v}}_k^{(t)\mathrm{T}}$, $t = 1, \cdots, T$.

---

**Algorithm 2** Deflation for gSTCCA

**Input:** Paired tensor data $\{\mathcal{X}_n, \mathcal{Y}_n\}_{n=1}^N$, CP-rank $R_u$ and $R_v$.
**Output:** Canonical tensor pair $\{\widehat{\mathcal{U}}_r\}_{r=1}^{R_u}$ and $\{\widehat{\mathcal{V}}_r\}_{r=1}^{R_v}$.
1: Initialize $\mathcal{U}^*$ and $\mathcal{V}^*$.
2: **while** not converged **do**
3:    **for** $r = 1, \cdots, R_u$ **do**
4:        Compute $\widehat{\mathcal{U}}_r$ in Eq. (15) by Algorithm 1,
5:        Compute the residual $\widehat{\mathbf{p}}_r$ by using Eq. (17).
6:    **end for**
7:    Update $\mathcal{U}^* = \sum_{r=1}^{R_u} \widehat{\mathcal{U}}_r$.
8:    **for** $r = 1, \cdots, R_v$ **do**
9:        Compute $\widehat{\mathcal{V}}_r$ in Eq. (16) by Algorithm 1,
10:       Compute the residual $\widehat{\mathbf{q}}_r$ by using Eq. (17).
11:   **end for**
12:   Update $\mathcal{V}^* = \sum_{r=1}^{R_v} \widehat{\mathcal{V}}_r$.
13: **end while**

---

### 4.3 Sequential Pursuit of Higher Rank

After the first pair of rank-one tensors is identified, it is desirable to pursue a higher-rank tensor canonical pathway for better performance. In the following, we show how more tensor components can be derived with a sequential rank-one tensor approximation algorithm based on the deflation technique (Chen et al., 2021). To streamline the presentation, let us assume that we can solve the rank-one problem in Eq. (6) to get $\widehat{\mathcal{U}}_r$ and $\widehat{\mathcal{V}}_r$ in each $r$-th step and select the tuning parameters $\lambda_u^r$ and $\lambda_v^r$ suitably, *e.g.*, cross validation. In general, the gSTCCA problem in the $r$-th step can be expressed alternatively as

$$\widehat{\mathcal{U}}_r = \arg\min_{\mathcal{U}_r} \frac{1}{2N}\|\mathcal{X} \times_1^S \mathcal{U}_r - \widehat{\mathbf{p}}_r\|_2^2 + \lambda_u^r\|\mathcal{U}_r\|_1,$$
$$\text{s.t. } Var(\mathcal{X} \times_1^S \mathcal{U}_r) = 1, Rank(\mathcal{U}_r) = 1, r = 1, \cdots, R_u, \tag{15}$$

$$\widehat{\mathcal{V}}_r = \arg\min_{\mathcal{V}_r} \frac{1}{2N}\|\mathcal{Y}\times_1^T\mathcal{V}_r - \widehat{\mathbf{q}}_r\|_2^2 + \lambda_v^r\|\mathcal{V}_r\|_1,$$

$$\text{s.t. } Var(\mathcal{Y}\times_1^T\mathcal{V}_r) = 1, Rank(\mathcal{V}_r) = 1, r = 1,\cdots,R_v, \tag{16}$$

where $\widehat{\mathbf{p}}_r$ and $\widehat{\mathbf{q}}_r$ are the current residues of canonical variables with

$$\widehat{\mathbf{p}}_r := \begin{cases} \mathcal{Y}\times_1^T\mathcal{V}^*, & \text{if } r = 1 \\ \widehat{\mathbf{p}}_{r-1} - \mathcal{X}\times_1^S\widehat{\mathcal{U}}_{r-1}, & \text{otherwise} \end{cases}, \tag{17}$$

$$\text{and } \widehat{\mathbf{q}}_r := \begin{cases} \mathcal{X}\times_1^S\mathcal{U}^*, & \text{if } r = 1 \\ \widehat{\mathbf{q}}_{r-1} - \mathcal{Y}\times_1^T\widehat{\mathcal{V}}_{r-1}, & \text{otherwise} \end{cases}, \tag{18}$$

where $\mathcal{U}^*$ and $\mathcal{V}^*$ are the initial values, $\widehat{\mathbf{p}}_{r-1}$ and $\widehat{\mathbf{q}}_{r-1}$ are the estimated values in the $(r-1)$-th step. Note that, the values $R_u$ and $R_v$ are not necessarily the same, since input data $\mathcal{X}$ and $\mathcal{Y}$ may differ in tensor structures.

It can be seen that the problems (15) and (16) are reduced to the same forms as given by Eqs. (7) and (8). Based on Algorithm 1, a deflation method for solving the optimization problems of gSTCCA in (15) and (16) that aims to find canonical tensor pairs $\{\widehat{\mathcal{U}}_r\}_{r=1}^{R_u}$ and $\{\widehat{\mathcal{V}}_r\}_{r=1}^{R_v}$, furthermore, is summarized in Algorithm 2.

## 5 Discussions

**Convergence Analysis.** It is proved that the SURF algorithm converges to a coordinate-wise minimum point (He et al., 2018). By construction of Steps 4 and 5 in Algorithm 1, we have $f(\widetilde{\mathcal{U}}_k,\lambda_{u,k}) \geq f(\widetilde{\mathcal{V}}_k,\lambda_{v,k}) \geq f(\widetilde{\mathcal{U}}_{k+1},\lambda_{u,k+1}) \geq f(\widetilde{\mathcal{V}}_{k+1},\lambda_{v,k+1})$. Thus, the iterative approach in Algorithm 1 (line 3 to line 7) monotonically decreases the objective function (6) in each iteration. Steps 8 and 9 are equivalent to constrained optimization problems in (9) and (10), which has been proved in Lemma 4.1 and is finite. Hence the number of iterations is finite in Algorithm 1. Moreover, Algorithm 2 repeatedly calls Algorithm 1 to generate sequential rank-one tensors based on the deflation, which has been proved that it leads to monotonically decreasing sequence (da Silva et al., 2015). Thus, Algorithm 2 monotonically decreases during deflation and converges in a finite number of steps.

**Complexity Analysis.** For simplicity, we assume that the size of the first view $\mathcal{X}^{P_1\times\cdots\times P_S\times N}$ is larger than that of the second view $\mathcal{Y}^{Q_1\times\cdots\times Q_T\times N}$. For each iteration, the most time-consuming part of Algorithm 2 lies in iteratively performing the tensor-vector multiplication and solving the rank-one tensor estimation by Algorithm 1. Based on the results in He et al. (2018), the computational complexity of gSTCCA per iteration is $O(NR_u\sum_{s\neq\hat{s}}^S(\prod_{j\neq s,\hat{s}}^S P_j))$. Since gSTCCA does not require explicit vectorization of input data, thus yielding the space complexity of $O(N\prod_{s=1}^S P_s)$.

**Orthogonal Constraints.** Unlike the standard CCA in Eqs. (3) and (4), for the gSTCCA model we do not enforce orthogonality, as the concurrency of sparsity and orthogonality constraints makes the problem much more difficult. On the other hand, sparsity and orthogonality have been two largely incompatible goals. For instance, as mentioned in Uematsu et al. (2019), a standard orthogonalization process such as QR factorization (Chan, 1987) will generally destroy the sparsity pattern of a matrix. Here we adopt the tensor orthogonality given in Definition 3.6 to investigate the influence of orthogonality constraints on our gSTCCA model. After solving gSTCCA in Algorithm 2, we can utilize the 'CP-ORTHO' framework (Afshar et al., 2017) to orthogonalize rank-one tensors (dubbed gSTCCA$_\perp$).

**Feature Projection.** To preserve the higher-order tensor structure of input data, after obtaining a sequence of canonical rank-one tensors $\{\mathcal{U}_r\}_{r=1}^{R_u}$, $\{\mathcal{V}_r\}_{r=1}^{R_v}$, we can project each pair of $\{\mathcal{X}_n,\mathcal{Y}_n\}_{n=1}^N$ onto the corresponding feature spaces spanned by factor matrices $\{\mathbf{U}^{(s)}\}_{s=1}^S$ and $\{\mathbf{V}^{(t)}\}_{t=1}^T$ as

$$\mathcal{X}_n^{new} = \mathcal{X}_n\times_1\mathbf{U}^{(1)}\times\cdots\times_S\mathbf{U}^{(S)},$$

$$\mathcal{Y}_n^{new} = \mathcal{Y}_n\times_1\mathbf{V}^{(1)}\times\cdots\times_T\mathbf{V}^{(T)}, \tag{19}$$

where $\mathbf{U}^{(s)} = [\mathbf{u}_1^{(s)},\cdots,\mathbf{u}_{R_u}^{(s)}] \in \mathbb{R}^{P_s\times R_u}$ and $\mathbf{V}^{(t)} = [\mathbf{v}_1^{(t)},\cdots,\mathbf{v}_{R_v}^{(t)}] \in \mathbb{R}^{Q_t\times R_v}$. The data pair $\mathcal{X}_n^{new}$ and $\mathcal{Y}_n^{new}$ are $S$-th order and $T$-th order tensors, respectively.

Table 1: Statistics of five real-world datasets in our experiments. We summarize the number of images, classes and sizes for both the original and preprocessed ones. We can see from it that our methods have no constraint on tensor input size.

| Type | Dataset | # Images | # Classes | Image size | Preprocessing | Size($\mathcal{X}$) | Size($\mathcal{Y}$) |
|---|---|---|---|---|---|---|---|
| Handwritten Digit | MNIST | 1000 | 3[*] | $28 \times 28$ | Cropped | $28 \times 14$ | $28 \times 14$ |
| Human Face | Yale | 165 | 15 | $32 \times 32$ | Wavelet | $32 \times 32$ | $16 \times 16$ |
| Brain Network | BP | 97 | 2 | $82 \times 82$ | fMRI/DTI | $82 \times 82$ | $82 \times 82$ |
| Facial Expression | JAFFE | 213 | 7 | $200 \times 180$ | Gabor | $200 \times 180$ | $23 \times 25 \times 40$ |
| Gait Sequence | Gait32 | 731 | 71 | $32 \times 22 \times 10$ | Optical flow | $32 \times 22 \times 10$ | $32 \times 22 \times 10$ |

[*] For a fair comparison, we use 3 classes instead of 10 classes and follow the same settings as Chen et al. (2021) to conduct experiments on the MNIST dataset.

Table 2: Visualization of original datasets with two views across five datasets.

## 6 Experiments

We evaluate the effectiveness and efficiency of gSTCCA through comprehensive experiments on diverse real-world tensor datasets. Our evaluation encompasses comparison with state-of-the-art methods in terms of classification accuracy, model sparsity, and training time. Furthermore, we delve into feature visualization, perform in-depth hyperparameter analysis, and examine the robustness of our approach in handling missing and noisy data. In all of our experiments, 80% of data are randomly selected for training, while the remaining 20% is used for testing. To mitigate randomness, we perform 10 independent cross-validation runs and report the average results. All experiments are run with MATLAB R2022b on a machine equipped with a 3.6GHz CPU and 16GB RAM.

### 6.1 Datasets

Table 1 summarizes the statistics of five real-world datasets used in our experiments. The original images with two different views are visualized in Table 2. For reproducibility purpose, more details of data description and preprocessing are provided as below.

**Data Preprocessing.  Handwritten Digit (MNIST).** MNIST database (LeCun et al., 1998) contains 60000 training and 10000 testing images of size $28 \times 28$ with labels from '0' to '9'. We choose a subset consists of 1000 images with labels '0', '1', and '2' and follow the same settings as Chen et al. (2021) to conduct our experiments. The goal is to learn correlated representations between the upper and lower halves (two views) of the original images.

**Human Face (Yale).** The face images are collected from the Yale database (Cai et al., 2007), which contains 165 images of size $32 \times 32$. For each image, we apply wavelet transformation to generate the corresponding encoded feature image using the *dw2* function in MATLAB. The original and encoded feature tensors are used as two different views.

Table 3: Properties of all compared methods. $\lambda_u$ and $\lambda_v$ are the regularization parameters, $R$ is the matrix/tensor rank, $K$ is the iteration number, $N$ is the sample size. For brevity we assume the size of $\mathcal{X}_n \in \mathbb{R}^{P_1 \times \cdots \times P_S}$ is larger than that of $\mathcal{Y}_n \in \mathbb{R}^{Q_1 \times \cdots \times Q_T}$ with tensor rank equal to $R$ for each view.

| Method | Input Type | Data Constraint | Regularizer | Hyperparameters | Time Complexity | Space Complexity |
|---|---|---|---|---|---|---|
| CCA (Hotelling, 1936) | Vector | — | — | $R$ | $O(\prod_{i=1}^{S} P_i^3)$ | $O(\prod_{i=1}^{S} P_i^2 + N \prod_{i=1}^{S} P_i)$ |
| SCCA (Avants et al., 2014) | Vector | — | $l_1(\mathbf{u}), l_1(\mathbf{v})$ | $\lambda_u, \lambda_v, R$ | $O(KR \prod_{i=1}^{S} P_i^2)$ | $O(\prod_{i=1}^{S} P_i^2 + N \prod_{i=1}^{S} P_i)$ |
| DCCA (Andrew et al., 2013) | Vector | — | — | Many* | $O(\prod_{i=1}^{S} P_i^3)$ | $O(\prod_{i=1}^{S} P_i^2 + N \prod_{i=1}^{S} P_i)$ |
| 3DCCA (Gang et al., 2011) | Tensor | $S = T$ | — | $R$ | $O(KN \sum_{i=1}^{S} P_i^3)$ | $O(N \prod_{i=1}^{S} P_i)$ |
| dTCCA (Chen et al., 2021) | Tensor | $S = T$ | — | $R$ | $O(KNR \sum_{i=1}^{S} P_i^3)$ | $O(N^2 + N \prod_{i=1}^{S} P_i)$ |
| STCCA (Wang et al., 2016) | Tensor | $S = T$ | $l_1(\mathbf{u}), l_1(\mathbf{v})$ | $\lambda_u, \lambda_v, R$ | $O(K^2 R \sum_{i=1}^{S} P_i^2 + KN \sum_{i=1}^{S} P_i^3)$ | $O(N \prod_{i=1}^{S} P_i)$ |
| spTCCA (Min et al., 2019) | Tensor | — | $l_1(\mathbf{u}), l_1(\mathbf{v})$ | $\lambda_u, \lambda_v, R$ | $O(\prod_{i=1}^{S} P_i^3 + \sum_{i=1}^{S} K P_i^3 R^3)$ | $O(\prod_{i=1}^{S} P_i^2 + N \prod_{i=1}^{S} P_i)$ |
| gSTCCA (Ours) | Tensor | — | $l_1(\mathcal{U}), l_1(\mathcal{V})$ | $\lambda_u, \lambda_v, R$ | $O(KNR \sum_{i \neq i}^{S} (\prod_{j \neq i,i}^{S} P_j))$ | $O(N \prod_{i=1}^{S} P_i)$ |

* "Many" means more than three hyperparameters.

**Brain Network (BP).** Brain networks play an important role in understanding brain functions. Bipolar disorder (BP) dataset (Whitfield-Gabrieli & Nieto-Castanon, 2012) is collected from two modalities, *e.g.*, functional magnetic resonance imaging (fMRI), and diffusion tensor imaging (DTI). We follow Liu et al. (2018) to preprocess the imaging data, including realignment, co-registration, normalization and smoothing, and then construct the brain networks from fMRI and DTI based on the Brodmann template, which are treated as two views.

**Facial Expression (JAFFE).** The JAFFE database (Lyons et al., 2020) contains female facial expressions of seven categories (neutral, happiness, sadness, surprise, anger, disgust and fear), and the number of images for each category is almost the same. We first crop each image to the size of $200 \times 180$, and then construct dataset of different sizes and orders. Specifically, we use the cropped image as the first view, and its 3D Gabor features as the second view.

**Gait Sequence (Gait32).** The Gait32 dataset (Lu et al., 2008) contains 731 video sequences with 71 subjects designed for human identification. The size of each gait video is $32 \times 22 \times 10$, which can be naturally represented by a third-order tensor with the column, row, and time mode. We calculate optical flow and obtain two 3rd-order tensors, which are used as two views (Wang et al., 2016).

## 6.2 Experimental Settings

**Baselines.** Table 3 summarizes the properties of all compared methods. Specifically, we investigate the viability of the proposed gSTCCA as feature extractor for classification tasks, and compare it with the following methods: standard CCA (Hotelling, 1936), sparse CCA (SCCA) (Avants et al., 2014), deep CCA (DCCA) (Andrew et al., 2013), three-dimensional CCA (3DCCA) (Gang et al., 2011; Lee & Choi, 2007), CP-based sparse tensor CCA (spTCCA) (Min et al., 2019), deflation-based tensor CCA (dTCCA) (Chen et al., 2021) and sparse tensor CCA (STCCA) (Wang et al., 2016). For STCCA we implement it by ourselves as the source code is not publicly available. For other compared methods, we use the source codes provided by the original authors. To obtain the corresponding feature representation for classification, for all tensor-based approaches, we use Eq. (19) to project each pair of input data $(\mathcal{X}_n, \mathcal{Y}_n)$ to get projected tensors $(\mathcal{X}_n^{new}, \mathcal{Y}_n^{new})$. Then $(\mathcal{X}_n^{new}, \mathcal{Y}_n^{new})$ are vectorized and concatenated to form the final feature vector $\mathbf{f}_n = [Vec(\mathcal{X}_n^{new}), Vec(\mathcal{Y}_n^{new})]$, which is further fed to the SVM classifier (Chang & Lin, 2011). For vector-based methods, we concatenate their extracted feature vectors as the input of SVM.

**Hyperparameter Settings.** For a fair comparison, the parameters of all compared methods are carefully tuned using the same 5-fold cross validation strategy on the training set based on classification accuracy. Specifically, rank $R$ is chosen from $\{5, 10, \cdots, 60\}$, for methods that enforce sparsity constraints, $\lambda_u$ and $\lambda_v$ are selected from $\{0.001, 0.005, \cdots, 0.1\}$. For DCCA, we select the number of hidden layers from 2 to 4 and learning rate $l_r$ from $\{0.001, 0.005, 0.01\}$.

Table 4: Performance comparison of different methods in terms of classification accuracy (mean ± std). 'n/a': results are not available due to method constraints.

| Methods / Datasets | CCA | SCCA | DCCA | 3DCCA | dTCCA | STCCA | spTCCA | gSTCCA$_\perp$ | gSTCCA |
|---|---|---|---|---|---|---|---|---|---|
| MNIST | 90.7 ± 1.4 | 97.1 ± 0.9 | 96.9 ± 1.1 | 96.2 ± 1.1 | 90.9 ± 3.8 | 97.3 ± 1.2 | 90.8 ± 4.2 | 97.1 ± 1.1 | **97.4 ± 1.3** |
| Yale | **89.0 ± 9.9** | **89.0 ± 7.9** | 84.6 ± 11.9 | 74.6 ± 16.2 | 75.0 ± 19.8 | 87.0 ± 8.2 | 73.6 ± 8.9 | 86.6 ± 12.3 | **89.0 ± 9.6** |
| BP | 61.1 ± 9.1 | 58.9 ± 10.8 | 51.1 ± 12.2 | 55.8 ± 11.9 | 55.0 ± 12.6 | 50.5 ± 10.6 | 52.1 ± 9.8 | 59.5 ± 13.6 | **63.2 ± 7.0** |
| JAFFE | 86.1 ± 7.2 | 86.3 ± 6.9 | 84.8 ± 6.8 | n/a | n/a | n/a | 81.5 ± 12.1 | 85.9 ± 12.2 | **90.0 ± 5.7** |
| Gait32 | 25.8 ± 6.8 | 26.6 ± 6.2 | 17.8 ± 4.3 | 35.4 ± 6.9 | 14.5 ± 4.8 | 34.6 ± 5.6 | 16.5 ± 3.1 | 25.8 ± 3.6 | **41.4 ± 5.2** |

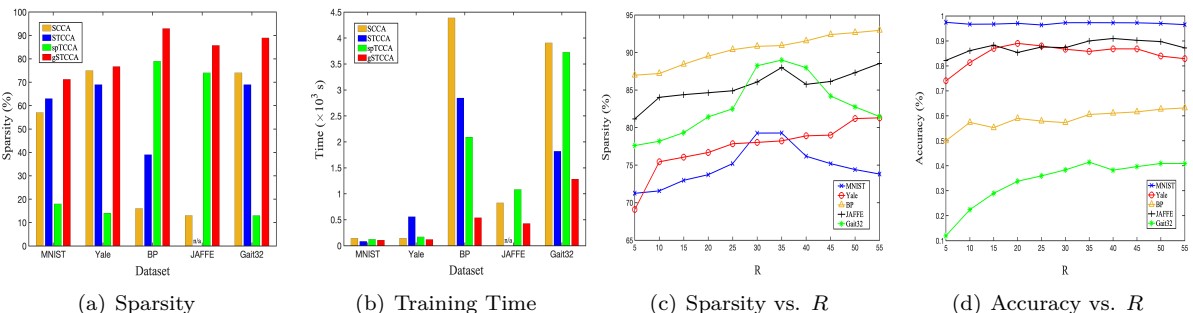

(a) Sparsity     (b) Training Time     (c) Sparsity vs. $R$     (d) Accuracy vs. $R$

Figure 1: Comparisons among sparse models. (a) and (b): Histogram of SCCA, STCCA, spTCCA and gSTCCA in terms of model sparsity and train time on five datasets. (c) and (d): Influence of $R$ on sparsity and accuracy of gSTCCA.

**Evaluation Metrics.** To quantitatively evaluate the performance of compared methods, we mainly use classification accuracy and model sparsity. The model sparsity is measured by the ratio of zero elements to the total number of elements on factor matrices $\{\mathbf{U}^{(s)}\}_{s=1}^{S}$ and $\{\mathbf{V}^{(t)}\}_{t=1}^{T}$.

## 6.3 Experimental Results

Table 4 presents the classification accuracy of all compared methods. The feature visualization of these compared methods on MNIST is shown in Fig. 2. The model sparsity and training time of the sparse approaches are illustrated in Figs. 1(a) and (b). Based on these results, we have the following observations: Overall, the proposed gSTCCA demonstrates very competitive performance in terms of accuracy, sparsity and efficiency. Specifically, gSTCCA produces the best results on all five datasets in terms of classification accuracy. The state-of-the-art STCCA method also shows outstanding performance, which reflects the advantage of pursuing sparse components. However, compared to STCCA, gSTCCA is more flexible since STCCA can not handle paired tensor data with different orders. Meanwhile, gSTCCA is also able to achieve higher sparsity, as can be seen from Fig. 1(a). In addition, we can see that gSTCCA outperforms gSTCCA$_\perp$ on all five datasets, indicating that the orthogonality constraints may introduce fitting noise and lead to poor approximation of the observed tensors. It is noticed that some related works also discarded orthogonality constraints of sparse components (Allen, 2012; Shen & Huang, 2008; Wang et al., 2016).

For efficiency, it is noticed from Fig. 1(b), gSTCCA is more computationally complex than STCCA, but it is also easy and efficient to train in that we can select the optimal $\lambda$ for $\mathcal{U}$ and $\mathcal{V}$ from their corresponding solution paths, thus eliminating the need to fine tune $\lambda_u$ and $\lambda_v$ from a pre-defined range. Besides, vector-based sparse method (*i.e.*, SCCA) is very fast when the feature size is small, while its training time grows significantly as the feature size increases. The reason is that the computational burden of SCCA lies mainly in performing singular value decomposition (SVD), which is time-consuming for large matrices. To summarize, the results above indicate that our gSTCCA is able to effectively and efficiently find high-quality sparse tensor coefficients $\mathcal{U}$ and $\mathcal{V}$ that capture the most useful components of input tensor data, and then boost the classification performance.

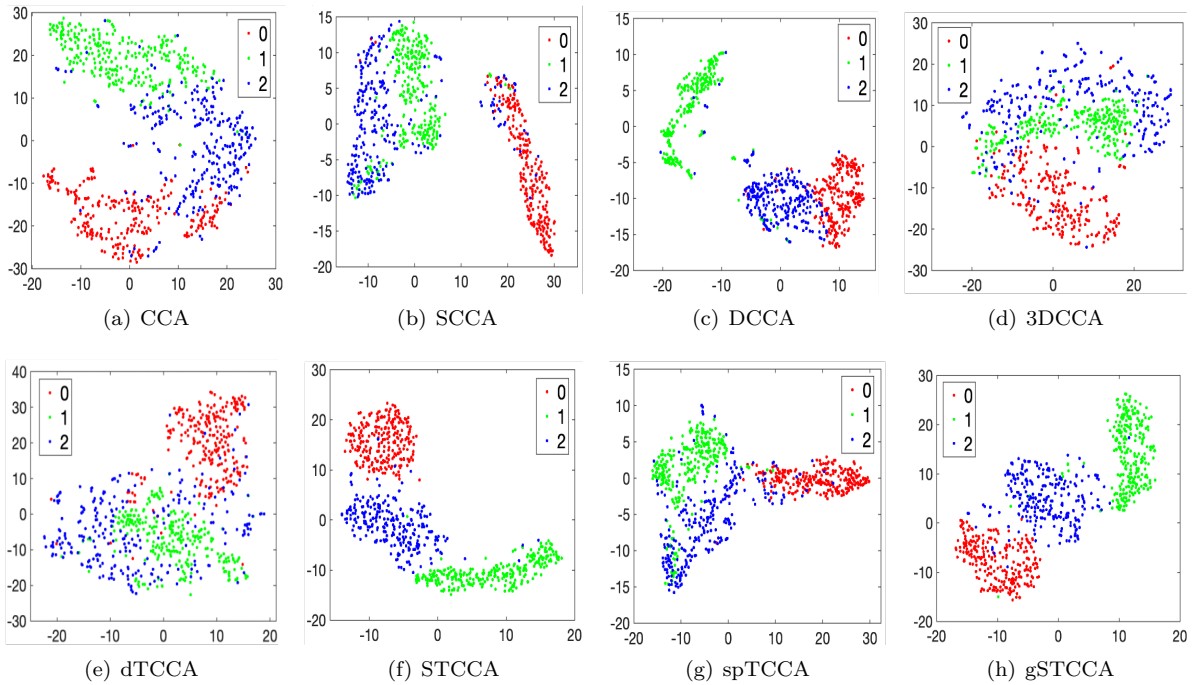

Figure 2: The t-SNE visualization of features obtained from compared methods on MNIST dataset.

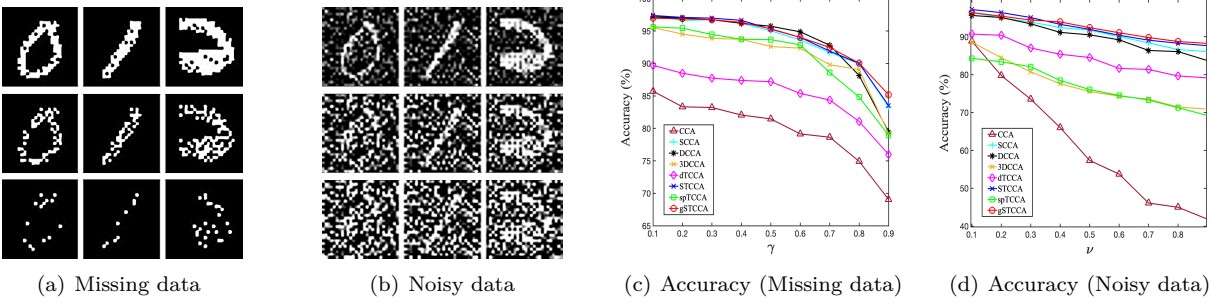

Figure 3: (a) and (b): Illustrations of MNIST images with missing ratio $\gamma \in \{0.1, 0.5, 0.9\}$ and noise variance $\nu \in \{0.1, 0.5, 0.9\}$. (c) and (d): Classification accuracy of all compared methods on missing and noisy data.

### 6.3.1 Feature Visualization

To qualitatively assess the effectiveness of gSTCCA, we use the t-SNE algorithm (Van der Maaten & Hinton, 2008) to visualize the features $\{\mathbf{f}_i\}_{i=1}^N$ learned by different methods on MNIST dataset. As shown in Fig. 2, the results indicate that only STCCA and gSTCCA can produce well-separated and compact clusters. However, from Table 4 in our paper, it is noticed that STCCA cannot handle paired tensor data with different orders.

### 6.3.2 Hyperparameter Sensitivity Analysis

The tensor rank $R$ is essential for tensor methods, since it directly controls the reduced dimensionality of feature space. Here we investigate the influence of tensor rank on the proposed gSTCCA over five datasets. For simplicity we assume $R = R_u = R_v$ and compare the sparsity and accuracy by varying $R$ from $\{5, 10, \cdots, 30\}$. From Fig. 1(c), we notice that the model becomes more sparse given a larger $R$, which can be explained

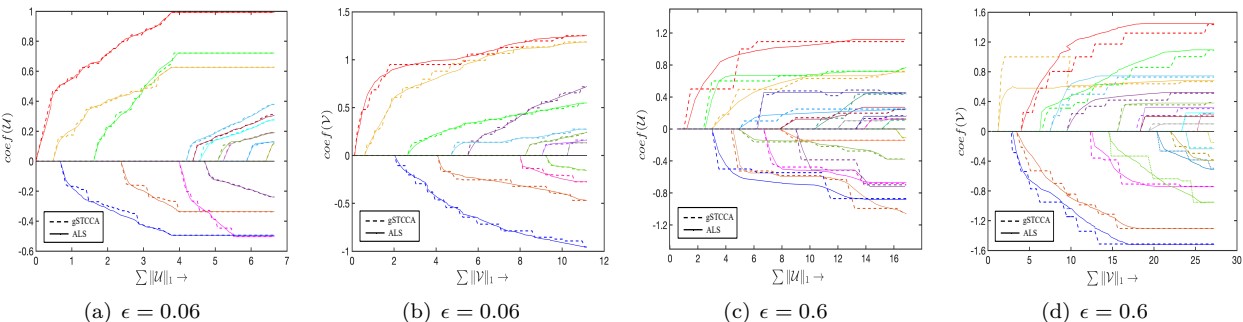

Figure 4: Comparison of solution paths of gSTCCA and ALS for a subset of canonical tensors $\mathcal{U}$ and $\mathcal{V}$ with small and large step sizes $\epsilon$ on MNIST dataset. The $x$-axis denotes the sparseness coefficient, and the $y$-axis denotes the value of the coefficients.

by the fact that important information tends to be contained in the first few tensor components. However, increasing $R$ does not always guarantee higher accuracy in that it may introduce some unwanted noise. For example, as can be seen from Fig. 1(d), the best results of gSTCCA on MNIST, Yale, BP, JAFFE and Gait32 are achieved with rank $R = 5, 20, 55, 10$ and $35$, respectively. In practice, this interesting observation may be useful to narrow the range of parameter tuning for related datasets and applications.

### 6.3.3 Influence of Missing and Noisy Data

In many real-world applications, images are inevitably contaminated by outliers such as missing data and noise during acquisition. Figs. 3(a) and (b) illustrate the MNIST data corrupted with certain missing ratio $\gamma$ and additive white gaussian noise (AWGN) of different variance $\nu$, respectively. We evaluate the robustness of all compared methods by varying $\gamma$ and $\nu$. From Figs. 3(c) and (d), it is noticed that the proposed gSTCCA is more robust to high missing data ratio and noise level, because the SVD adopted by compared methods may be more sensitive to extreme outliers.

### 6.3.4 Solution Path

Fig. 4 shows the collection of solution paths of gSTCCA for a subset of canonical tensors on MNIST dataset, under both big and small step sizes. The path of estimates $\mathcal{U}$ and $\mathcal{V}$ for each $\lambda$ is treated as a function of $t = \sum \|\mathcal{U}\|_1$ and $t = \sum \|\mathcal{V}\|_1$, respectively. The $x$-axis denotes the sparseness coefficient, and the $y$-axis denotes the value of the coefficients. The vertical lines denote (a subset of) the turning point of the path, as the solution path for each of the coefficients is piece-wise linear. From Fig. 4, we can see that the gSTCCA path is almost indiscernible from that of ALS when the step size is small, which matches well with the theoretical analysis in He et al. (2018).

### 6.3.5 Application on ADNI

To show the significance and benefits of integrating tensor-structured sparsity into the CCA framework, we perform the task of predicting the stage of Alzheimer's Disease (AD) using real-world data sourced from the Alzheimer's Disease Neuroimaging Initiative (ADNI) database[2]. The dataset comprises 116 features extracted from brain regions of interests (ROIs) across three medical imaging modalities: Voxel-Based Morphometry Magnetic Resonance Imaging (VBM-MRI), [18]F-fluorodeoxyglucose Positron Emission Tomography (FDG-PET), and 18-F florbetapir PET (AV45-PET). The feature grouping is performed using the MarsBaR toolbox Tzourio-Mazoyer et al. (2002). For detailed information on data acquisition and preprocessing, please refer to Yu et al. (2022).

---

[2]www.adni-info.org

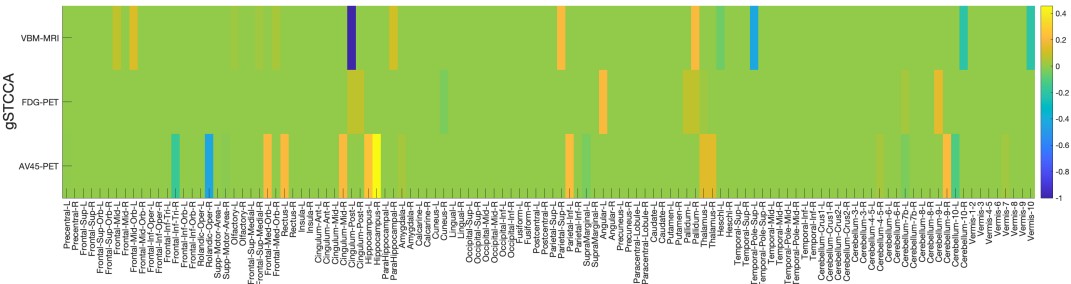

Figure 5: Visualization of coefficient weights on the ADNI data. From top to bottom, three rows are corresponding to three imaging modalities: (i.e., VBM-MRI, FDG-PET, and AV45-PET). From left to right, there are 116 brain regions defined in Tzourio-Mazoyer et al. (2002). We can easily find that gSTCCA can pay relatively balanced attention to all three modalities. The top 10 brain regions selected by our method are: Cingulum-Post-L, Hippocampus-R, Hippocampus-L, Cingulum-Mid-R, Frontal-Med-Orb-L, Rectus-L, Temporal-Pole-Sup-L, Rolandic-Oper-R, Angular-L, Parietal-Inf-L. Most of these selected regions are known to be highly related to AD and Mild Cognitive Impairment (MCI) in previous studies Du et al. (2001); Miklossy (2011); Vlassenko et al. (2012); Mattis et al. (2016); Bubb et al. (2018)

## 7    Conclusion

In this paper, we propose a novel sparse tensor CCA model for arbitrary tensor data. Specifically, we formulate the problem as a constrained multilinear least-squares problem with tensor-structured sparse regularization, and propose a divide-and-conquer strategy based on the deflation technique to sequentially solve the problem. This solution results in a set of unconstrained linear least-squares problems which can be efficiently solved. Extensive experiments conducted on five different datasets demonstrate the efficacy and robustness of our approach in terms of classification accuracy, model sparsity and having missing and noise data. Our future work will extend the proposed gSTCCA to multi-view learning and deep learning. It is also interesting to explore how to better fuse gSTCCA with orthogonality constraints, which can help to discover distinct and easily interpretable patterns from tensor data (Afshar et al., 2017).

## Acknowledgments

This work is partially supported by the NSF grants (MRI-2215789, IIS-1909879, IIS-2319451, DMS-2053697), NIH grants (1R01LM014344, 1R01AG077820, R01LM012607, R01AI130460, R01AG073435, R56AG074604, R01LM013519, R56AG069880, U01TR003709, RF1G077820, R21AI167418, R21EY034179), and Lehigh's grants under Accelerator and CORE. We also would like to acknowledge and thank all the individuals who shared their source code and datasets, or made them publicly available.

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
