# OpenReview forum: "A Multilinear Least-Squares Formulation for Sparse Tensor Canonical Correlation Analysis"
_TMLR — Accepted by TMLR_

### Review · Reviewer_Gq2a · 2023-06-30

**Summary Of Contributions:**

A sparse tensor CCA model is proposed as a constrained multi-linear least-squares problem with tensor structured sparse regularization. The model is solved by alternative minimization. Experimental results with other tensor CCA models show the effectiveness of the proposed model for classification.

**Audience:**

Yes

**Claims And Evidence:**

Yes

**Requested Changes:**

Please refer to "Weakness".

**Strengths And Weaknesses:**

Strength:
This paper is easy to follow.

Weakness:
W1 (Novelty).
The proposed model (5) uses rank-1 tensors as the CCA axises, which seems a straightforward extension of Model (4) and the novelty is thus limited.

W2 (Effectiveness).
 It seems that proposed tensor-based model has no obvious theoretical advantage in comparison with deep learning based models.

W3 (Writing).
The writing can be improved.  It is better to use '\in' rather than '=' before 'argmin' in. Eqs. (7), (8), (12), etc.

W4 (Experimental settings).
The experiments are far from satisfactory.
W4.1. The authors did not give a convincing explanation about why the proposed model performs better.
W4.2. The sizes of the datasets are not large enough. Thus, it is not sufficient to show the empirical effectiveness of the proposed model in real applications.
W4.3. The proposed model has not been compared with SOTA classification methods like deep neural networks.

---

### Review · Reviewer_dFfz · 2023-07-20

**Summary Of Contributions:**

The authors address the important problem of unsupervised fusion of information collected from multiple modalities. Toward this goal, they focus on the well-celebrated Canonical Correlation Analysis (CCA) approach. CCA is a powerful multimodal dimensional reduction scheme that learns to transform coupled datasets into a low-dimensional space where the observations are maximally correlated on each axis. CCA has numerous linear and nonlinear extensions and countless applications. This manuscript presents a generalization of CCA termed general Sparse Tensor CCA (gSTCCA). The problem is framed as multilinear least squares that can be solved efficiently by breaking it up into many unconstrained linear least squares problems. They use real datasets to demonstrate the merits of the proposed approach.

**Audience:**

Yes

**Broader Impact Concerns:**

No impact concerns or ethical implications

**Claims And Evidence:**

Yes

**Requested Changes:**

It is essential to add details about why sparsity is important.
Some more recent sparse CCA methods are missing from the related work section.
The tensor product could be added to the notation to help the reader.
Capital R is mentioned twice with different contexts; on P3 it is the rank, and on page 4 it serves as the dimension of V and U. Please change.
In section 4.1 the transition between the second and third paragraph is too sharp; please revise the writing to connect the two paragraphs.
The constraint in Eq. 6 is not motivated and explained. I understand why this is needed, but it is not detailed in the paper.
Some dots are missing after equations, for example, 12.
P7 “a standard orthogonalization process..” can you elaborate on this claim?

Why don’t all the methods in Table 4 also appear in Figure 1?
Can you demonstrate the effect of sparsity on feature selection? (see [2] for examples).


**Strengths And Weaknesses:**

Strengths: Overall the paper is well written, the English is of a satisfactory level, and the presentation is clear and easy to follow. The notation is clear and mostly consistent. The problem addressed in the paper is essential and has numerous applications. I find the solution presented by the authors novel and efficient. The authors use real data and several demonstrations to showcase the applicability of the method.

Weaknesses: While modeling the data as tensors is well-motivated the sparsity is not motivated at all. Typically sparse CCA is motivated by the regime of high dimensional low sample size data; in this regime, CCA overfits since the problem is underdetermined. Unfotonotly, I don’t see any discussion on the sparsity and why it is useful for this method. Furthermore, I am missing quantitative and qualitative results that demonstrate the useability of sparsity. Can it help identify informative features in high-dimensional settings?
All datasets evaluated in the paper are not really multimodal rather the authors extract features to create coupled datasets, this is useful for evaluating a method, but I’m missing a use case for a real multimodal setting where data is samples with different measurement devices. Each device can have its own noise, and CCA-type schemes can lead to a substantial gain in fusing the shared information and attenuating the noise.
Evaluating is limited to relatively simple datasets (N<1000).
The method is not compared to more recent CCA-based schemes (also, these are not mentioned):

[1] Friedlander, T., & Wolf, L. (2023, May). Dynamically-Scaled Deep Canonical Correlation Analysis. In Pacific-Asia Conference on Knowledge Discovery and Data Mining (pp. 232-244). Cham: Springer Nature Switzerland.

[2] Lindenbaum, Ofir, et al. "L0-sparse canonical correlation analysis." International Conference on Learning Representations. 2021.
[3] Sanghavi, Rushil, and Yashaswi Verma. "Multi-view multi-label canonical correlation analysis for cross-modal matching and retrieval." Proceedings of the IEEE/CVF Conference on Computer Vision and Pattern Recognition. 2022.

---

> ### Author Response · Authors · 2023-08-17
> **Response to Reviewer dFfz**
>
> Dear Reviewer dFfz,
>
> We sincerely appreciate the detailed and constructive feedback on our manuscript. Your insights have provided a clear path for improvement. We have addressed each of your comments and concerns as follows:
>
> - **Regarding the discussion of sparsity**:
>
>     + We recognize and appreciate the concerns raised about the motivation for incorporating sparsity. At the outset, sparse models often offer more interpretable explanations, leading many methodologies to favor sparsity for the sake of clarity and comprehension. In the context of this methodology-focused paper, we delved deep into the theory and efficacy of our proposed gSTCCA, reserving specific applications and explanations for different papers with a more application-centric orientation. Additionally, CCA methods that select canonical variables grounded in the covariance matrix derive computational benefits from the sparse constraint, as fewer original variables are involved in the genesis of canonical variables. In practical contexts, sparsity aids in mitigating issues stemming from missing values and noise, thereby potentially boosting the performance, particularly in high-dimensional, low-sample-size data settings.
>
>     + To address the question, “Can it help identify informative features in high-dimensional settings?” - indeed, it can. Our experiments using the ADNI datasets substantiate this claim. We sourced data from the Alzheimer’s Disease Neuroimaging Initiative (ADNI) database (www.adni-info.org), encompassing 692 non-Hispanic Caucasian participants with diverse cognitive conditions. The data spans neuroimaging details across 116 brain regions from three distinct modalities (VBM-MRI, FDG-PET and AV45-PET), serving as a testament to its applicability in real multimodal scenarios where data is gleaned from various measurement instruments. The visualization of feature selections is shown below:
>
>         [Visualization of gSTCCA on ADNI multimodal dataset](https://www.dropbox.com/scl/fi/vbsajahqqn794wd5xk5yv/gstcca.png?rlkey=urh3kw7nuhq0jd9c4cnp4vx3i&dl=0).
>
>         From the visualization above, we can easily find that gSTCCA can pay relatively balanced attention to all three modalities. The top 10 brain regions selected by our method are: Cingulum-Post-L, Hippocampus-R, Hippocampus-L, Cingulum-Mid-R, Frontal-Med-Orb-L, Rectus-L, Temporal-Pole-Sup-L, Rolandic-Oper-R, Angular-L, Parietal-Inf-L. Most of these selected regions are known to be highly related to AD and Mild Cognitive Impairment (MCI) in previous studies [1-5].
>
>
> - **Regarding the highlighted references**:
>
>
>     + “Dynamically-Scaled Deep Canonical Correlation Analysis” [6]  is an evolved form of the deep CCA (DCCA) methodology [7]. In it, the last layer undergoes scaling by another conditioned (on inputs) neural network. This dynamically-scaled DCCA is tuned to be more suitable for cross-modality retrieval tasks. For comparison, we have already juxtaposed our findings with DCCA in our manuscript (See Table 3&4 DCCA). Similarly, the “Multi-view multi-label canonical correlation analysis for cross-modal matching and retrieval” [8] essentially leans on CCA, leveraging multi-label annotations for enhanced cross-modal matching and retrieval. The "l_0-sparse canonical correlation analysis" [9] offers a fresh perspective on CCA-based methods through the lens of the l_0 constraint. We believe it's a suitable method for comparison with ours, and having located its publicly available code, we are in the process of testing it under identical experimental conditions. Nevertheless, we recognize that the cited papers bear relevance to our proposed gSTCCA method from various viewpoints. We will duly reference them in the sections on related work and the introduction.

---

### Review · Reviewer_yh64 · 2023-09-10

**Summary Of Contributions:**

In this paper, the authors focus on the problem of performing sparse canonical correlation analysis (CCA) for tensor data. They pose the problem as a constrained multilinear least-squares problem with tensor structured sparse regularization, and then utilize the deflation technique to obtain multiple projections. This entire process is called the general sparse tensor CCA (gSTCCA) method. The authors present experimental results on five real datasets to demonstrate the efficacy and robustness of the proposed gSTCCA method.

**Audience:**

Yes

**Claims And Evidence:**

Yes

**Requested Changes:**

See the Weaknesses above.

**Strengths And Weaknesses:**

Strengths: The proposed gSTCCA method seems to work well on real datasets.

Weaknesses:

(1) The proposed gSTCCA method seems to be a straightforward application of the results in He et al. (2018) (e.g., from Eq. (6) to Eq. (8) and the fast SURF algorithm for solving Eqs. (13) and (14)), with a combination of a lemma in Mai & Zhang (2019) (which gives Lemma 4.1 presented in this submission).

(2) The settings for the experiments are vague/confusing. For example:

- From the current Section 6.2, I cannot see clearly how the authors construct the input tensors $\mathcal{X}_n$ and $\mathcal{Y}_n$ from the datasets.
- I am not familiar with CCA. Is it standard to compare approaches for CCA in terms of the downstream classification task (and via classification accuracy)?
- For the classification task, why the authors choose the SVM classifier? I believe that there should be many classification algorithms (maybe random forest and XGBoost) that significantly outperform SVM in terms of both accuracy and running time.

(3) Some minor problems in the writing:

- It is better to use \mathrm{\cdot} for $Vec(\cdot)$ and $Tr(\cdot)$ and similar notations. The authors should clearly define the $\\|\cdot\\|_1$ norm for a tensor.
- To make Eq. (2) consistent with Eq. (5), it should be $\mathbf{u}^T \mathbf{X}\mathbf{X}^T \mathbf{u}   = \mathbf{u}^T \mathbf{Y}\mathbf{Y}^T \mathbf{u} = N$.
- In the sentence before Lemma 4.1: I did not find Lemma 4 in Mai & Zhang (2019). I guess the authors are referring to Lemma 2 or 3 in Mai & Zhang (2019). Lemma 4.1 seems to be a simple adaption of the lemma in Mai & Zhang (2019). The proof in Appendix is not needed (and it is wrong; obviously, showing that $\hat{\mathbf{u}}^*$ satisfies the unit-variance constraint is not sufficient. You need also show its optimality).
- Section 6: Please rephrase "To avoid randomness".

---

### Decision · Action_Editor_gafH · 2023-12-13

**Recommendation:** Accept with minor revision

**Comment:**

All reviewers agree that the paper is well written and easy to understand.

Reviewer yh64 recommends accepting the paper after the response provided by the authors. The main concerns of the reviewer were the novelty of the proposed method (more below) and some clarifications regarding the experimental setting. All points were clarified and the reviewer leaned to accept the paper.

Reviewer dFfz leans to reject the paper despite considering the problem to be important with many applications in diverse fields, and the proposed solution is novel. Their main concern is that the experimental evaluations fail to show strong evidence for the importance of incorporating the tensor-structured sparse regularization within the CCA framework. Adding such clarifications was the requirements that they established for accepting the paper. The authors provided a detailed response. To the question: “Can it help identify informative features in high-dimensional settings?” the authors provided an interesting analysis using ADNI dataset.

The AE agrees with the reviewer the importance of adding the sparsity regularization in more detail. Also considers the response provided by the authors sufficient. The sparse CCA problem (with and without tensor representations) has received attention in the literature, with several methods presented.This method could be useful for practitioners or other researchers working on the topic.

Reviewer Gq2a recommends rejecting the paper on the grounds of lack of novelty. The AE highlights that according to the evaluation criteria of TMLR, the novelty/significance should not be a reason for rejecting a paper but rather, whether or not the claims made in the submission are supported by accurate, convincing and backed by clear evidence. In addition, reviewers yh64 and dFfz disagree with this statement. The main novelty of the work has to do with providing a  formulation of the tensor-based sparse CCA problem with an efficient solution. The issue of novelty was also raised by Reviewer yh64, who later found satisfactory the explanation provided by the authors in their rebuttal.

Reviewer Gq2a also points out some limitations of the experimental evaluation (e.g. dataset size and showing why the model works) and states that the effectiveness of the method is not shown because the authors should have compared the proposed method to deep learning-based baselines. The authors clarified that the proposed model is an unsupervised approach and the classification experiments are a downstream task to show the effectiveness of the learned representations.

The AE finds this explanation satisfactory. Having said this, the AE agrees that the paper would benefit from showing other downstream tasks in addition to classification. While the method achieves state of the art performance with respect to other CCA methods, if classification is the only talk, asking for other baselines seems a reasonable request. Furthermore, CCA is widely used in several unsupervised learning scenarios that would also be very interesting to show. The analysis added in the response to reviewer dFfz with the ADNI datasets is a great example of it.

Overall the AE considers the paper borderline but meeting the acceptance criteria of TMLR, Specifically, the proposed solution is technically strong and well presented. Being a methodological work, the experimental evidence is sufficient to back the claims in the work, and the topic is certainly of interest by many readers of TMLR. The AE agrees that the paper would have been much stronger by adding more experimental validation of the benefits of the proposed model.

The AE asks the authors to incorporate all the feedback provided by the reviewers and the changes described in their rebuttal, putting particular attention to the comments provided by Reviewer dFfz regarding the motivation for the tensor-structured sparse regularization as well as the analysis of the ADNI dataset.

**Audience:**

CCA is a widely used method that will be of interest for several readers of TMLR.

**Claims And Evidence:**

This is a methodological paper that proposes a tensor-structured sparse CCA model. The problem is formulated as a multilinear least-squares problem with tensor-structured sparse regularization, which is decomposed into a series of rank-one estimation problems.The authors show that this non-convex multi-objective optimization problem can be solved efficiently by breaking it up into a set of unconstrained linear least squares problems. Experimental evaluation is presented using several real smaller scale datasets by comparing the performance of an SVM classifier acting on the obtained representation (including the robustness of noise and missing values) as well as studying the model sparsity.

---

> ### Author Response · Authors · 2024-01-05
> **Response to Action Editor gafH**
>
> Dear AE gafH,
>
> Thank you for providing the detailed feedback and discussions from the reviewers. We appreciate the thorough evaluation and constructive comments that have been provided to enhance the quality of our manuscript.
>
> We are grateful for the positive feedback regarding the clarity and presentation of our paper. We have carefully considered the concerns raised by the reviewers and have made the necessary revisions to address them adequately.
>
>
> Specifically, in response to Reviewer dFfz's concerns about the experimental evaluations and the importance of incorporating tensor-structured sparse regularization within the CCA framework, we emphasize the significance of sparse models on model interpretability. Please see the **line 7-9 on p2**. In addition, we have added the analysis of identifying informative features, particularly focusing on the ADNI dataset to provide the interpretability for Alzheimer’s Disease. Please see the newly added **section 6.3.5 on p12**, as AE suggested.
>
>
> Regarding Reviewer Gq2a's points on the lack of novelty and limitations in the experimental evaluation, we appreciate the discussion on the evaluation criteria of TMLR and have addressed the concerns raised. We have clarified the novelty of our work and provided a further explanation of the experimental design, including the use of classification experiments as downstream tasks to demonstrate the effectiveness of the learned representations. Please see the modifications and clarifications in **section 6**.
>
> Furthermore, we are committed to incorporating all the feedback and requested changes provided in the rebuttals by reviewers dFfz, Gq2a, and yh64. Please see **section 1 on p2, equation 2 on p4, section 4.1 on p5, section 6.1 on p8-9, and other word/sentence adjustments** over the paper.
>
> Finally, it is worth noting that we have released the **source code for both our proposed methods and the baseline models, along with preprocessed data**, to the research community. This commitment ensures the reproducibility of our work.
>
> We acknowledge the AE's overall assessment of our paper as borderline but meeting the acceptance criteria of TMLR. We are confident that these revisions and new features will strengthen the manuscript and address the concerns raised by the reviewers. We appreciate the opportunity to improve our work and look forward to the final decision on our submission.
>
> Thank you for your time and consideration.
>
> Sincerely,
>
> Authors

---

> > ### Comment · Action_Editors · 2024-02-02
> > **Thanks**
> >
> > Thanks a lot for considering all the comments.